# Validation of CloudSat-CPR Derived Precipitation Occurrence and Phase Estimates across Canada

Rithwik Kodamana and Christopher G. Fletcher *

Department of Geography and Environmental Management, University of Waterloo,
Waterloo, ON N2L 3G1, Canada; rithwik.kodamana@uwaterloo.ca
* Correspondence: chris.fletcher@uwaterloo.ca

**Abstract:** Snowfall affects the terrestrial climate system at high latitudes through its impacts on local meteorology, freshwater resources and energy balance. Precise snowfall monitoring is essential for cold countries such as Canada, and particularly in temperature-sensitive regions such as the Arctic; however, its size and remote location means the precipitation gauge network there is sparse. While satellite remote sensing of snowfall from instruments such as CloudSat-CPR offers a potential solution, satellite detection of precipitation phase has not been systematically evaluated across Canada. In this study, CloudSat-based precipitation occurrence and phase retrievals were validated at 26 stations across Canada maintained by Environment and Climate Change Canada (ECCC). Probability of Detection (POD), defined as the percentage agreement between coincident CloudSat and human-observed present weather information for precipitation (solid, liquid or no precipitation), and False Alarm Ratio (FAR) were used as the primary metrics for validation. The mean POD (FAR) for precipitation occurrence across Canada is 65.5% $\pm$ 4.3 (31.4% $\pm$ 5.1) and for no precipitation is 90.6% $\pm$ 1.4 (11% $\pm$ 2.5). The results show lower rates of detection under cloudier skies, in the presence of (freezing) drizzle and for lighter snowfall, which may be explained by a large number of false-positives due to CloudSat-CPR's high instrumental sensitivity. When CloudSat correctly detects the occurrence of precipitation, it shows uniformly high POD (>80%) and low FAR (<10%) for classifying the *phase* of precipitation. Large databases of coincident ground and satellite measurements allow us to provide a new estimate of around 9% for the frequency of virga events, a factor of two smaller than a previous estimate for the Arctic. The results from this study show that CloudSat has useful accuracy in detecting precipitation occurrence and very high accuracy at classifying precipitation phase, over diverse climate zones across Canada. As such, there is significant potential for satellite monitoring of snowfall in remote, cold regions.

**Keywords:** remote sensing; CloudSat; POSS; snowfall; arctic; precipitation phase; ground validation





## 1. Introduction

Snow is an important component of the global climate system and cryosphere, playing integral roles in Earth's water and energy balance [1,2]. Snow covers approximately 47 million sq. km on average of Northern Hemisphere land each year [3,4]. Snowfall readily changes the surface temperature, impacts atmospheric dynamics and circulation patterns and affects permafrost extent [5,6]. Snowfall is particularly important to the cultural identity and economy of Canada through its influence on short-term weather and long-term climate by altering the surface energy budget, local interactions with wind and temperature in the Arctic and other snow-covered regions, freshwater storage, tourism and transportation [7–9]. Therefore, precise snowfall monitoring is essential for cold countries such as Canada. Environment and Climate Change Canada (ECCC) maintains a national network of 1735 surface weather stations, but station density is low in remote regions such as the Canadian Arctic due to challenges related to access and climate [10]. Snowfall is known to be highly variable in space and time, and thus the Canadian gauge network may be too sparse to obtain reliable snowfall estimates over an entire region [11,12].

Another significant challenge which influences the accuracy of snowfall monitoring is the identification of precipitation phase. Different phases of precipitation affect land hydrology and climate differently [13]. Misclassification of precipitation phase causes substantial errors in the estimates of Snow Water Equivalent (SWE), snow depth, snowfall rate, snow albedo feedback and streamflow [14–17]. While gauges measure accumulated precipitation, the observation of precipitation phase is often carried out by trained human observers who report present weather information or by automated sensors [18,19]. Human involvement helps to distinguish between weather types, some of which are not detectable by automated instruments [19]. However, human observations of precipitation phase at high latitudes are made very challenging by the long periods of darkness during the polar night [20].

Satellite remote sensing provides the potential to overcome several of these challenges by providing global or quasi-global measurements of frozen precipitation [21]. Passive microwave sensors have been used to estimate snow properties across Canada with moderate success [22,23]; however, passive microwave retrievals tend to underestimate low-intensity precipitation at higher latitudes [24,25] and suffer from coarse spatial resolution [12,26]. Relative to passive sensors, active microwave sensors have higher spatial resolution, and active radar instruments such as the Cloud Profiling Radar (CPR) aboard the NASA Cloud-Sat satellite provide information on the vertical structure of clouds and precipitation [27]. CloudSat-CPR was the first instrument to provide active space-borne snowfall observations on a near-global scale, including in remote, high-latitude regions [28,29]. The near-daily high latitude coverage, combined with the high sensitivity of the CPR (94 GHz), offer great potential for snowfall research in the Arctic [30]. However, several instrumental and environmental factors impact CloudSat's ability to retrieve surface snowfall accurately. The CPR instrument is designed to retrieve cloud properties, rather than precipitation, which means that water vapor, liquid and frozen hydrometeors may attenuate the CloudSat radar retrieval [31]. Surface clutter, due to radar pulses interacting with the ground surface, is another significant source of contamination in CloudSat retrievals within the boundary layer [29], with twice the rate of false hydrometeor detection below 2 km altitude than above [32].

Despite these challenges, recent validation of CloudSat-estimated surface precipitation phase has shown promising results. Hudak et al. [33] evaluated CloudSat's precipitation occurrence algorithm with C-Band ground weather radar between September 2006 and April 2007 at King City, Ontario, Canada and reported a Probability of Detection (POD) value of 94.7%. Norin et al. [6] compared CloudSat snow occurrences with Swedish weather radar network (SWERAD) data from January 2008 to December 2010 and reported POD values of 60–90% for precipitation intensities <0.1 mm/h. Chen et al. [34] obtained a POD value of 76.1% for CloudSat snow occurrences in comparison with the NOAA–MRMS product over CONUS from January 2009 to December 2010. However, since ground-based radar is also retrieval-based, it naturally introduces additional sources of uncertainty. To our knowledge, a detailed validation of CloudSat's retrieval of precipitation occurrence over Canada has not been performed against in situ observations (automated or human observed), and it is necessary to provide a retrieval-independent evaluation of CloudSat's snow detection algorithm.

The primary goal of this work is to validate precipitation phase estimates from Cloud-Sat using ground observations across Canada, with a particular focus on high latitude remote locations such as the Canadian Arctic. We also quantify whether CloudSat's accuracy varies for different weather types and precipitation intensities. Section 2 introduces the datasets used in this study, along with details of the data processing and validation methodology. Section 3 provides the validation results for one station in Eureka, Nunavut, while Section 4 describes the pan-Canada results. Finally, Section 5 provides a discussion of the essential findings, limitations and future research.

## 2. Datasets and Methodology

### 2.1. ECCC Hourly Present Weather Observations

The "ground-truth" data in this study are collected from 26 ECCC weather stations across Canada (Figure 1), which provide observations of solid, liquid and non-precipitating weather types (referred to simply as 'ECCC weather data' for the rest of this paper). This subset of ECCC stations represents the locations where a sufficiently large sample of coincident ground observations and CloudSat overpasses are available. The observations are recorded at each station once per hour, on the hour, by trained human observers [19]. The different solid, liquid and non-precipitating weather types from the ECCC weather data that are included in this study, and their frequencies, are shown in Table 4. We explicitly exclude precipitation types such as ice crystals, which are produced by boundary layer clouds [20], due to known contamination of near-surface CloudSat retrievals by surface clutter [32].

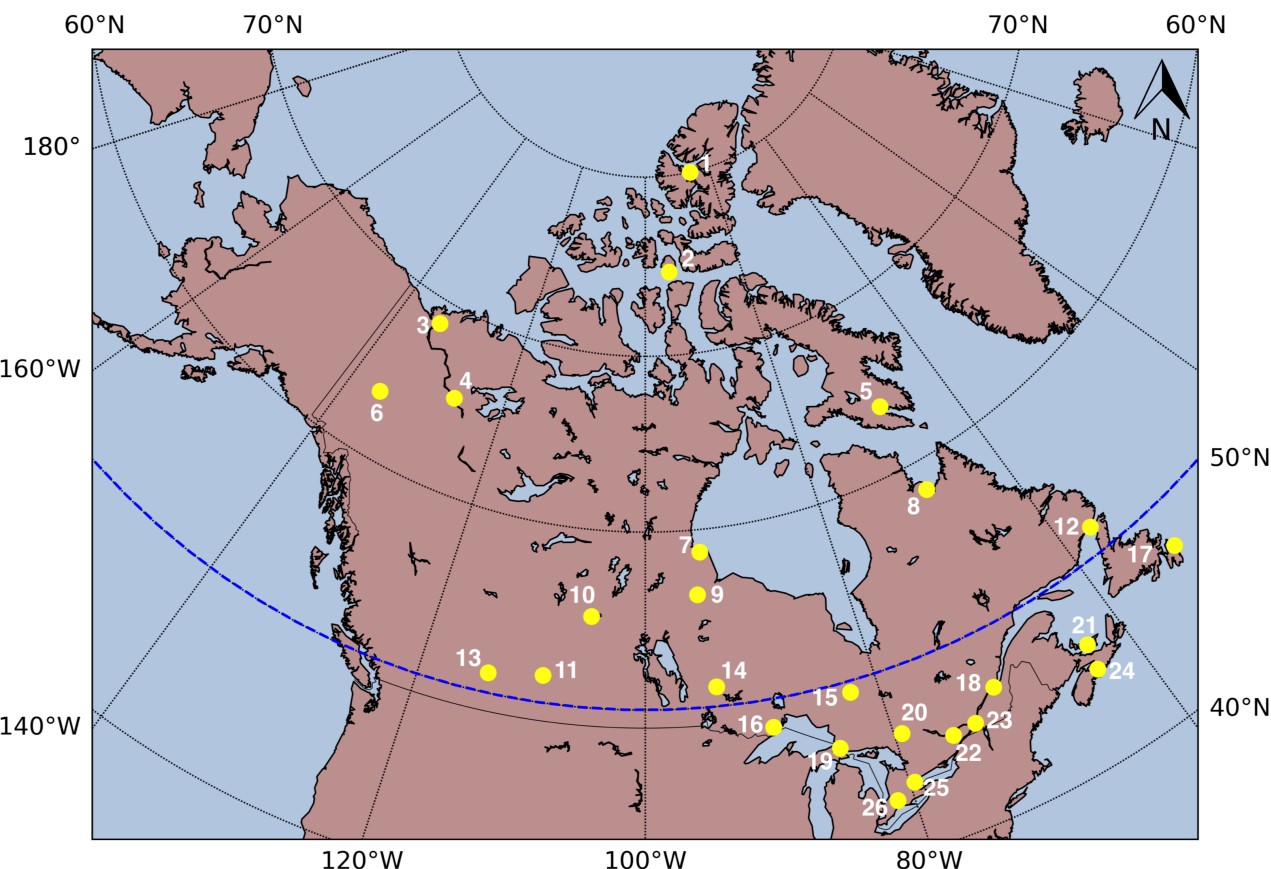

**Figure 1.** Distribution of ECCC weather stations used in this study as listed in Table 3. The dotted blue line shows the 50° N parallel, which separates northern from southern stations in the pan-Canada validation.

### 2.2. CloudSat-CPR

NASA launched CloudSat in 2006 carrying the 94 GHz Cloud Profiling Radar (CPR) [35]. Every 0.16 s along CloudSat's sun-synchronous orbital track, CPR sends radar pulses into the atmosphere below and receives the backscattered power to form a vertical profile over 125 discrete layers (each 240 m thick, and referred to as "bins") extending from the ground surface to 30 km altitude [32,36]. A collection of many such vertical profiles along the instrument's orbital track is referred to here as an overpass, and we describe a methodology below to examine collocated overpasses over time at locations close to ECCC ground stations. Table 1 shows the CloudSat data products used in this study.

The preliminary CloudSat product that identifies the occurrence of precipitation, and its phase, is 2C-PRECIP-COLUMN [37]. Following the identification of precipitation oc-

currence, a decision tree with temperature and reflectivity thresholds is used to classify the phase of precipitation [37]. CloudSat's classification of precipitation phase is based on contemporaneous meteorological information from the ECMWF-AUX operational analysis, which provides the set of ancillary state variables, such as atmospheric temperature, interpolated to the same vertical bins as CloudSat [38,39]. For each CloudSat profile, precipitation is classified as snow when surface temperature (T) T < 0 °C, rain when T > 2 °C or mixed when 0 < T < 2 °C [37]. Next in the retrieval process, for profiles identified as snow, the radar reflectivity value for the bin nearest to the ground surface is combined with the assumed particle size distribution and microphysical parameters to estimate the snowfall rate at the surface, which is provided in the 2C-SNOW-PROFILE product.

**Table 1.** CloudSat data products and extracted variables used in this study.

| CloudSat Product | Version | Extracted Variables | Units |
|---|---|---|---|
| ECMWF-AUX | P_R05 | 2 m temperature | K |
| 2C-PRECIP-COLUMN | P_R05 | Precipitation flag | - |
| | | Melted mass fraction | - |
| | | Near surface reflectivity | dBZe |
| | | Height of top of lowest significant cloud layer | km |
| 2C-SNOW-PROFILE | P1_R05 | Surface snowfall rate | mm/h |

Ground clutter causes CloudSat retrievals to be unreliable over complex surface topography [40]. We mitigate against the effects of ground clutter by following the established practice of masking the five vertical bins nearest to the surface [28]. While this ensures that the reflectivity features being examined are free from the interference of surface topography, the masking procedure means that no retrieval information is available from the lowest 1200 m of the atmosphere. This gives rise to the well-known "blind zone" in CloudSat's retrievals and means that surface snowfall rates over land are approximated using the snowfall rate in the *sixth* vertical bin (approximately 1500 m altitude) [41,42]. This creates an important source of uncertainty in comparisons between space-based and ground-based measurements of precipitation, because precipitation formed within the planetary boundary layer by shallow convection is very likely to be missed by CloudSat [30], and precipitation that evaporates or sublimates within the boundary layer (virga) may be undetected at the surface.

### 2.3. POSS Weather Data

The Precipitation Occurrence Sensor System (POSS) is a ground-based, upward-looking X-band radar that provides an estimate of precipitation occurrence, phase and intensity (mm h$^{-1}$) at a temporal resolution of one minute [43,44]. POSS is mounted 3 m above the ground and measures the Doppler signal of falling hydrometeors through a small sampling volume with maximum size limited to 1 m$^3$ above the sensor. The instrument uses the measured Doppler signal to estimate the present weather type and precipitation intensity [45]. Owing to the higher temporal sampling rate of POSS (1 min) compared to ECCC present weather observations (1 h), the POSS instrument provides an independent verification of the influence of temporal sampling in our method of CloudSat validation. In this paper, we include POSS observations from the instrument located at the ECCC weather station at Eureka, NU.

### 2.4. Method of Validation

In this study, precipitation occurrence and phase information from CloudSat overpasses transiting within a 100 km spatial radius centered on each ECCC ground station (Figure 2a) are compared to coincident observations from the ground station. We classify a CloudSat profile's surface precipitation phase as solid when its near-surface precipitation flag indicates either *snow possible* or *snow certain* and as liquid when its precipitation flag

indicates *rain possible*, *rain probable* or *rain certain*. The individual profile classifications are then aggregated to assign an overall precipitation phase for each overpass using the weights of solid ($\hat{w}_s$) and liquid ($\hat{w}_l$) profiles contained in each overpass. Each profile's weight is estimated as its inverse distance (1/distance) from the ECCC weather station under consideration. To be considered as a precipitating overpass, the sum of weights of liquid and solid precipitation profiles in an overpass must exceed a threshold of 30% of the total weight of individual profiles in the same overpass. The set of overpasses not satisfying this criteria are classified as non-precipitating. The threshold of 30% was selected following a sensitivity analysis, which demonstrated that 30% provides a reasonable balance in sample size of precipitating and non-precipitating overpasses (not shown). Precipitating overpasses are classified as solid precipitation when $\hat{w}_s > \hat{w}_l$ and as liquid precipitation when $\hat{w}_l > \hat{w}_s$.

The snowfall rate for each overpass is computed as the mean of the surface snowfall rate of all individual solid precipitation profiles in the overpass. In our analysis, mixed precipitation profiles (identified when the CloudSat precipitation flag indicates *mixed possible* or *mixed certain*) are classified as either solid or liquid based on the value of the melted mass fraction, which is the mass fraction of snow that has undergone melting. The CloudSat snow estimation algorithm considers mixed precipitation occurrences with melted mass fraction values $\leq 0.15$ as snow, and a surface snowfall rate is estimated for these overpasses [46]. The same threshold is used in this study for the conversion of mixed precipitation occurrences to solid (melted mass fraction $\leq 0.15$) or liquid (melted mass fraction $> 0.15$).

Additionally, to test the robustness of the inverse distance weighting (IDW) method, we also present results using a "Default" method, which classifies overpasses as precipitating or non-precipitating based only on the number of precipitation profiles in an overpass. In Default, an overpass is classified as precipitating when it contains *at least one* profile flagged by CloudSat as solid or liquid and as non-precipitating when the overpass does not contain *any* solid or liquid precipitation profiles. A precipitating overpass is classified as solid or liquid based on the proportion of solid ($\hat{p}_s$) and liquid ($\hat{p}_l$) precipitation flags it contains. The classification is solid when $\hat{p}_s > \hat{p}_l$ and liquid when $\hat{p}_s \leq \hat{p}_l$. The Default method sets a highly stringent threshold for identifying non-precipitating overpasses; i.e., if *any* profiles in an overpass indicate precipitation, then the overpass will be classified as either liquid or solid, regardless of how many profiles indicate precipitation or where they are located in relation to the weather station.

The Probability of Detection (POD), False Alarm Ratio (FAR) and Heidke Skill Score (HSS) metrics are used to quantify how often CloudSat correctly detects the occurrence of precipitation, and its phase, relative to ground observations [47]. To ensure a robust comparison, we group sub-types of each precipitation category for the ground observations (Table 4) into the solid, liquid or no precipitation categories used by CloudSat. This means that the three types of solid precipitation recorded by ECCC observers are combined into a single group called 'solid precipitation' [13,17]. To provide an illustration for the case of solid precipitation, the metrics are estimated using the following equations based on the legend provided in Table 2. The POD (also known as the 'hit rate') is defined as the percentage of events where CloudSat and the ground observation agree on either precipitation occurrence (precipitating or non-precipitating) or precipitation phase (solid or liquid):

$$\text{POD} = \frac{a}{a+c}. \tag{1}$$

The FAR is defined as the frequency of 'false positive' measurements by CloudSat:

$$\text{FAR} = \frac{b}{a+b}. \tag{2}$$

**Table 2.** Legend for comparing CloudSat and ground observations. In this example, a and d refer to the frequency of correct classification by CloudSat for solid and liquid precipitation, respectively. b refers to misclassification where the ground observation was liquid, but CloudSat classified solid precipitation and c is the reverse. A similar table (not shown) is used for precipitating and non-precipitating events.

| CloudSat \ Ground | Solid | Liquid |
|---|---|---|
| Solid | a | b |
| Liquid | c | d |

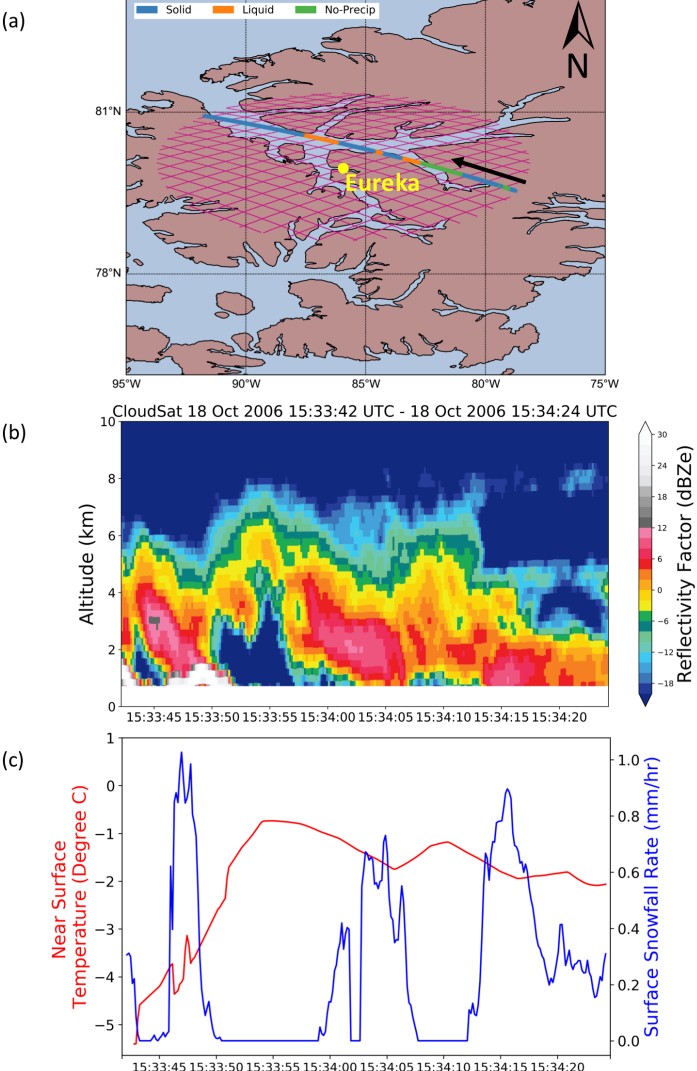

**Figure 2.** Figures demonstrating the details of the overpass validation methodology for a mixed precipitation event at Eureka, Nunavut recorded on 18 October 2006 at UTC 15:33. (**a**) A spatial radius of 100 km centered on Eureka weather station, and the magenta lines indicate the tracks of CloudSat overpasses included in this study. Each individual overpass comprises retrieval profiles recorded along the track every 0.16 s. The multicolored line shows one CloudSat overpass on 18 October 2006, with retrieval profiles color-coded by precipitation occurrence and type using the method described in Section 3.1. (**b**) The vertical profile of reflectivity measured by CloudSat along the overpass track. (**c**) The coincident time series of near-surface temperature from the CloudSat ECMWF-AUX product (red curve) and surface snowfall rate from CloudSat 2C-SNOW-PROFILE (blue curve) for the same overpass. Surface snowfall rates of zero indicate profiles coded either as no precipitation or as rain.

An example of a false alarm is where CloudSat detects precipitation but none is observed on the ground (or vice versa). Additionally, the Heidke Skill Score (HSS) [48] combines POD and FAR information into a single metric to describe the accuracy:

$$\text{HSS} = \frac{2(ad - bc)}{(a + c)(c + d) + (a + b)(b + d)}. \tag{3}$$

Our validation procedure assumes that the weather type recorded on the ground remains constant for a time $\tau$, where $\tau$ is the sampling frequency of each ground-based dataset: $\tau_{ECCC}$ = 1 h and $\tau_{POSS}$ = 1 min [6,19,34]. The temporal matching of each Cloud-Sat overpass with a ground-based weather observation is carried out by restricting the maximum time difference between the two observations to $\tau/2$. The impact of temporal sampling on our results is tested in Section 3.2.

## 3. Validation at Eureka, NU

The primary ground station used in this analysis is Eureka, Nunavut (WEU), which, due to its high latitude location (80° N), benefits from a very high density of CloudSat overpasses available from July 2006 to December 2016. Eureka is the ideal site for CloudSat validation, because a large database of coincident measurements is available from CloudSat, ECCC and POSS weather sensors.

### 3.1. Detection of Precipitation Occurrence and Phase

To illustrate how our precipitation detection, classification and validation methods work, Figure 2a highlights a single overpass near Eureka weather station from 18 October 2006 at UTC 15:33. A precipitating weather system was passing close to the weather station at the time, and the color of each dot in the overpass corresponds to the precipitation flag recorded by CloudSat (liquid, solid or no precipitation). This overpass includes profiles that are flagged as liquid, solid and no precipitation, and those indicating precipitation are associated with increased reflectivity in the level closest to the surface in Figure 2b.

Near-surface air temperatures at the time of this overpass are very close to freezing (Figure 2c), causing melting of the frozen hydrometeors as they approach the surface, and the presence of both liquid and solid precipitation profiles. At approximately UTC 15:33:45, a region with enhanced reflectivity at 3 km (maximum ~12 dbZ) decreases in altitude as the satellite moves northwest along its track (Figure 2b). The slight reduction in reflectivity around 2 km altitude, along with the coincident increase in temperature, likely indicates melting and disintegration of snowflakes [49,50]. The region with a sharp reduction in reflectivity centered at UTC 15:33:55 is associated with an area of non-precipitation. At 15:34:10, we see a region of higher reflectivity, increased temperature and zero surface snowfall rate (Figure 2c), suggesting the complete conversion of snowflakes to raindrops coinciding with a change in precipitation flag from solid to liquid (Figure 2a).

The weight of CloudSat retrievals with precipitating and non-precipitating flags for this overpass are 0.82 and 0.18, respectively. Since the weight of precipitation profiles are greater than 0.3, this overpass is recorded as having detected precipitation. The weights of solid and liquid precipitating profiles for this precipitating overpass are $\hat{w}_s$ = 0.53 and $\hat{w}_l$ = 0.29, respectively. As $\hat{w}_s > \hat{w}_l$, this overpass is classified as a solid precipitation event. The present weather observation recorded at the time closest to the CloudSat overpass (UTC 16:00) indicates 'snow' as the precipitation type received on the ground, and therefore we record this as a successful detection by CloudSat (a 'hit'). As the different types of solid precipitation reported by ECCC observers are consolidated into a single group 'solid precipitation', the result would also be considered a 'hit' even if the ground observation had reported another type of solid precipitation (e.g., snow grains). If the precipitation phase of the CloudSat overpass does not agree with the matched ECCC present weather observation, then that overpass would be recorded as a 'miss'. In the same manner, for each ECCC ground station, each overpass with a matched present-weather observation in

our database is recorded as a hit or miss. The counts of hits and misses are used to estimate the POD, FAR and HSS values at each station.

Expanding this methodology to all 3052 matched CloudSat–ECCC observations at Eureka (Table 3), we find POD (FAR) values of 62% (46.4%) and 88% (9%) for precipitating and non-precipitating conditions, respectively, and the HSS is 0.47 (Figure 3a). Among the subset of precipitating observations correctly detected by CloudSat, we find POD (FAR) values of 98% (4%) and 78% (5%) for solid and liquid precipitation, respectively (Figure 3c). The HSS value is 0.83. The results suggest that CloudSat is prone to missing the occurrence of precipitation and exhibits higher accuracy in classifying the phase of precipitation once it is detected. Although our results compare favourably with previous estimates of POD and FAR for CloudSat-detected precipitation (e.g., [34]), it is striking that around 40% of precipitation events are recorded as misses. In the next section, we explore the primary factors that influence errors in detection by CloudSat.

**Table 3.** Table showing the geographical coordinates of stations considered in this study as shown in Figure 1, station codes and number of solid, liquid and non-precipitating CloudSat overpasses (sample size) obtained after matching CloudSat and ECCC weather data

| Sl No. | Station | Station Code | Lat (°) | Lon (°) | Solid | Liquid | No-Precip |
|--------|---------|--------------|---------|---------|-------|--------|-----------|
| 1 | Eureka | WEU | 79.99 | −85.93 | 502 | 81 | 2469 |
| 2 | Resolute−Bay | YRB | 74.72 | −94.97 | 391 | 85 | 762 |
| 3 | Inuvik | YEV | 68.67 | −133.68 | 163 | 86 | 637 |
| 4 | Norman Wells | YVQ | 65.28 | −126.80 | 153 | 54 | 617 |
| 5 | Iqaluit | YFB | 63.75 | −68.54 | 113 | 55 | 511 |
| 6 | Mayo | YMA | 63.62 | −135.87 | 88 | 55 | 632 |
| 7 | Churchill | YYQ | 58.73 | −94.07 | 95 | 30 | 342 |
| 8 | Kuujjuaq | YVP | 58.34 | −68.38 | 166 | 103 | 499 |
| 9 | Gilllam | YGX | 56.34 | −94.70 | 86 | 42 | 418 |
| 10 | La Ronge | YVC | 55.11 | −105.29 | 83 | 33 | 416 |
| 11 | Kindersley | YKY | 51.52 | −109.18 | 33 | 29 | 428 |
| 12 | Blanc Sablon | YBX | 51.44 | −57.13 | 55 | 79 | 405 |
| 13 | Calgary | YYC | 51.11 | −114.02 | 39 | 29 | 389 |
| 14 | Red Lake | YRL | 51.09 | −93.69 | 64 | 40 | 196 |
| 15 | Kapuskasing | YYU | 49.40 | −82.41 | 94 | 63 | 410 |
| 16 | Thunder−Bay | YQT | 48.45 | −89.32 | 44 | 38 | 341 |
| 17 | St. John's | YYT | 47.62 | −52.74 | 40 | 86 | 262 |
| 18 | Quebec | XBO | 46.83 | −71.25 | 53 | 66 | 397 |
| 19 | Sault Ste Marie | YAM | 46.57 | −84.41 | 61 | 54 | 338 |
| 20 | North−Bay | YYB | 46.40 | −79.39 | 79 | 58 | 325 |
| 21 | Charlottetown | YYG | 46.29 | −63.13 | 34 | 48 | 249 |
| 22 | Montreal | YUL | 45.47 | −73.74 | 44 | 49 | 374 |
| 23 | Ottawa | YOW | 45.32 | −75.67 | 27 | 46 | 355 |
| 24 | Halifax | YHZ | 44.88 | −63.51 | 35 | 75 | 345 |
| 25 | Toronto | YYZ | 43.68 | −79.63 | 17 | 42 | 313 |
| 26 | London | YXU | 43.00 | −81.25 | 39 | 43 | 256 |
| | pan-Canada | | | | 2598 | 1469 | 12,686 |

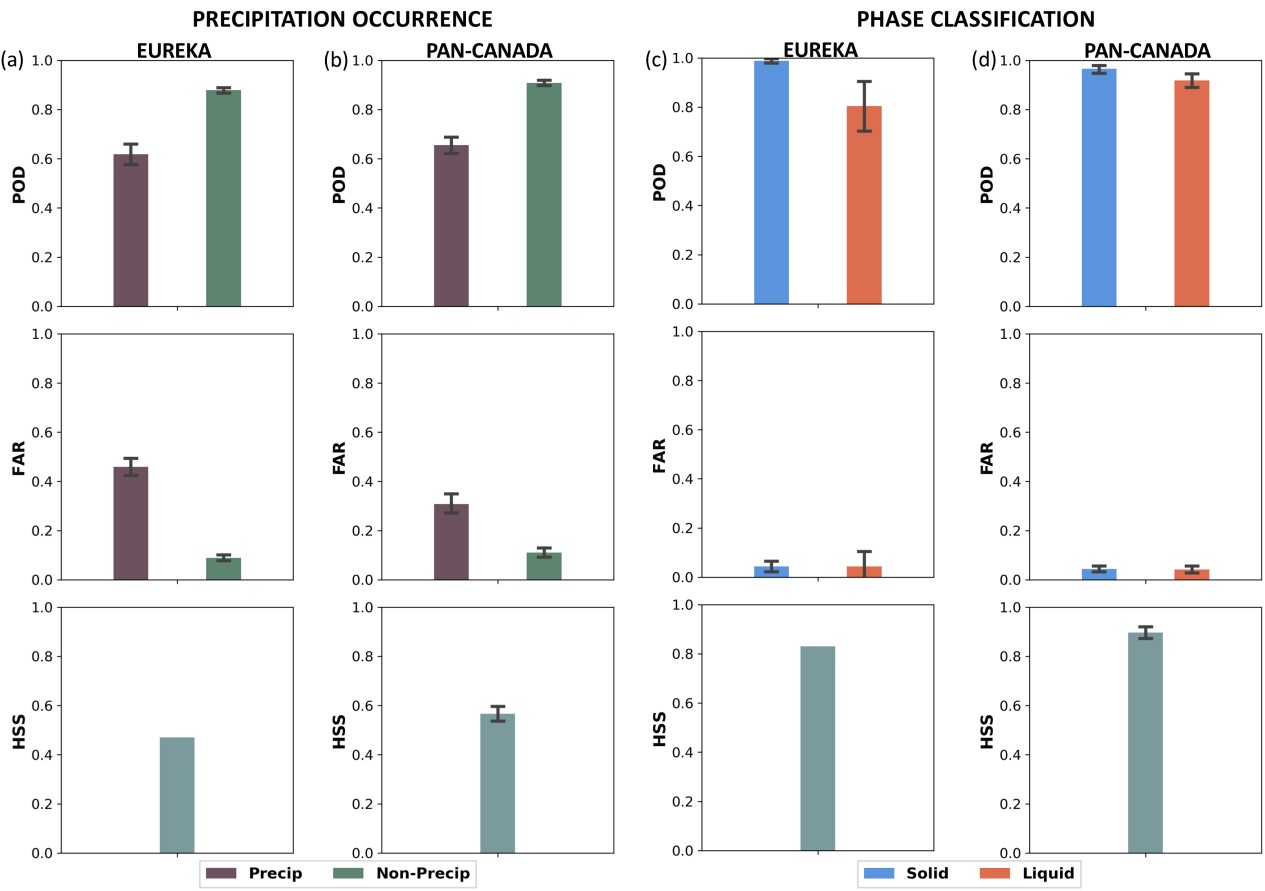

**Figure 3.** Estimates of POD (top), FAR (middle) and HSS (bottom) at Eureka (**a**,**c**) and pan-Canada (**b**,**d**) for precipitation occurrence (**a**,**b**) and phase classification (**c**,**d**). The error bars represent 95% confidence intervals on each estimate, assuming that the samples are *i.i.d.*

### 3.2. Factors Influencing Detection

Five main sources of uncertainty likely contribute to the misclassification of precipitation by CloudSat: mismatches in the (i) sampling, (ii) spatial location and (iii) timing, between the ground observation and the CloudSat overpass; (iv) shallow precipitation generated within the planetary boundary layer; and (v) differences in sensitivity between human observers and the CPR instrument on board CloudSat. In this section, we investigate these different effects to assess what are the dominant factors controlling the rate of missed detection.

Figure 4 shows the results of a sensitivity analysis performed at Eureka to quantify the influence of spatiotemporal sampling on our results. We define the Sample Ratio (SR) as the percentage of the available matched overpasses at Eureka that are included; for example, $SR = 10$ indicates that the analysis is based on a random sample containing just 10% of the original data. By repeating the resampling 100 times, we seek to quantify the uncertainty in precipitation detection that exists at less well-sampled stations. For example, when $SR = 10$, the available sample size of 338 is similar to that of London, ON (Table 3), one of the lowest in our dataset. The skill score (HSS) is remarkably stable across a range of SR values (Figure 4a), with a slight narrowing of the uncertainty range at higher SR, indicating that our results are robust even at the lowest latitude stations with only a few hundred observations.

Next, we also test the impact on HSS from varying the distance from Eureka station over which CloudSat overpasses are included in the analysis. There is a small decline in HSS with increasing spatial radius (Figure 4b), but in general the accuracy is stable above 70 km when using our IDW method (Section 2). In contrast, when no distance weighting is applied to the CloudSat profiles, there is a larger decrease in HSS above 50 km,

indicating that the Default method is more susceptible to sampling uncertainty. This result demonstrates the importance of applying distance weighting. Using a 100 km radius approximately doubles the available sample size relative to a 50 km radius, and so we made the decision to use the 100 km radius to maximize data availability.

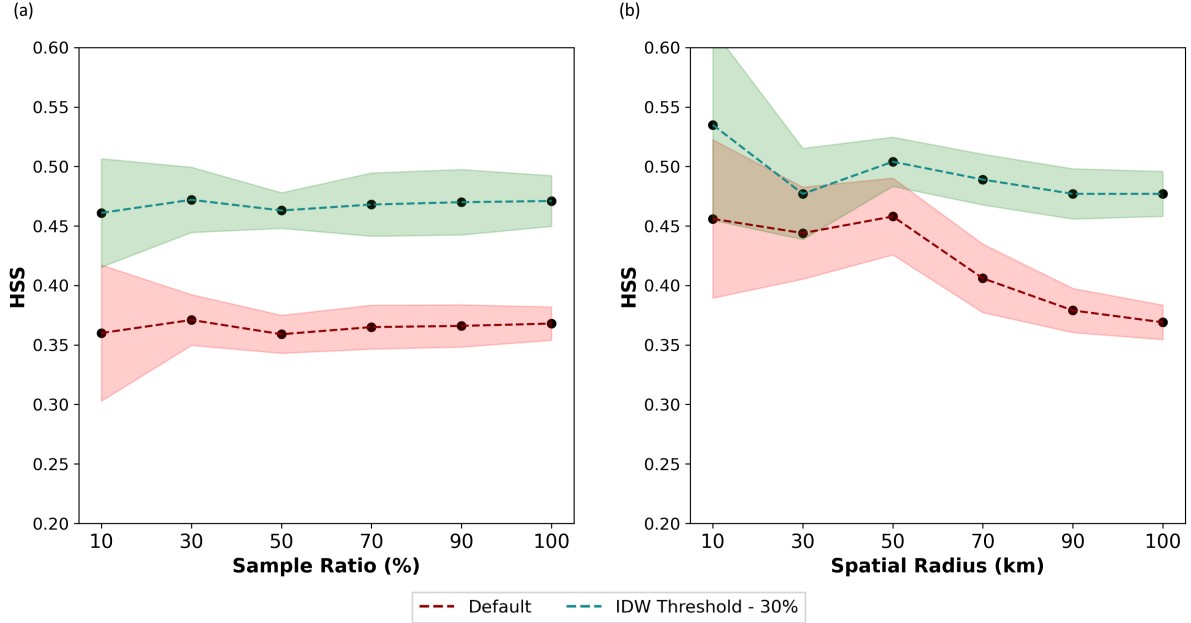

**Figure 4.** Sensitivity analysis showing how HSS for precipitation occurrence at Eureka varies as a function of: Sample Ratio (%) (**a**); and Spatial Radius (km) (**b**). The red dashed line shows results from the Default method, while the green dashed line shows results that include inverse distance weighting (IDW). The shaded region indicates the uncertainty in each estimate based on a $n = 100$ bootstrap resampling, with replacement (see Section 3.2 for more details).

To investigate what fraction of missed detections is caused by the mismatch in *timing* of up to ±30 min between the satellite overpass and the ground observation, we computed POD, FAR and HSS values using the POSS observations at Eureka, which are available within ±30 s of the satellite overpass. Interestingly, the POD, FAR and HSS values calculated using POSS are similar to those reported above for ECCC ground observations both in detecting the occurrence of precipitation and classifying the phase of precipitation (not shown). POSS is a ground-based, upward-looking radar, and so another useful property of the instrument is that it can detect shallow precipitation occurring within the boundary layer, inside the CloudSat blind-zone. The similarity in POD when using CloudSat and POSS suggests that the temporal sampling does not have a major impact on the misclassification rate of precipitation phase estimated by CloudSat.

Due to its high Arctic location, Eureka experiences complete darkness during winter months and constant daylight during summer. One might expect that nighttime conditions would make it challenging for even a trained human observer to correctly observe precipitation phase, particularly during the extremely cold winter months, and that this could introduce a bias in our validation toward lower POD during polar night. To investigate this, we crudely separate the data at Eureka into day and night observations by grouping them by season: winter (October to February) and summer (March to September). The POD values for precipitating and non-precipitating conditions during winter months are 62.0% and 85.0%, respectively, and the equivalent values during summer months are 61.5% and 89.6%, respectively. In other words, the POD is similar during predominantly day and night conditions at Eureka. This suggests that day/night conditions do not affect the accuracy of human observations, and the day–night cycle (or extreme cold temperatures) is not playing a major role in the rate of missed detection.

Next, we investigate CloudSat's accuracy of detection and classification under different weather conditions. The CloudSat POD for observations detecting no precipitation decreases with increasing cloud cover, ranging from 97% under clear skies to 70% under full cloud (Figure 5a). This is explained by the reflectivity threshold (−15 dBz) used to flag precipitation occurrence in the CloudSat 2C-PRECIP-COLUMN product [51], since radar backscatter received from a cloud layer may occasionally be above threshold even if hydrometeors are not reaching the surface. In the very cold climate of Eureka, snow is the dominant form of precipitation throughout the year, and the ECCC data record contains very few observations of snow grains or "moderate snow", which denotes heavier snowfall rates of 1–5 mm h$^{-1}$ (Table 4). Only around 2% of the ground observations at Eureka show liquid precipitation, and of those observations the only category with a sample size larger than $n = 10$ is rain (Table 4). Therefore, we cannot determine the influence of different types and sizes of hydrometeors on CloudSat's POD at Eureka, but a much larger sample of events is available for pan-Canada, which is discussed in Section 4.

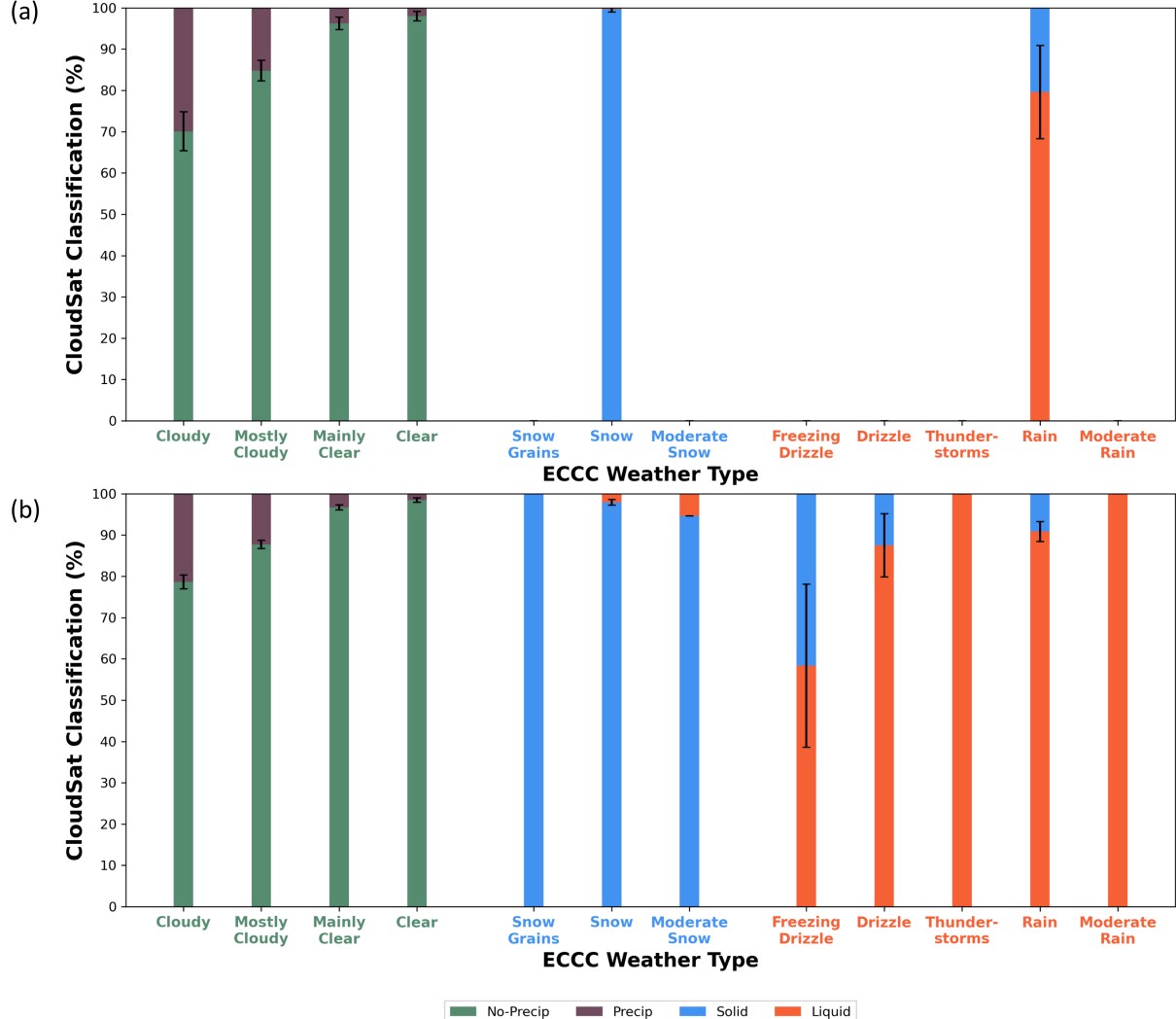

**Figure 5.** Detection of precipitation occurrence and classification of its phase, by CloudSat (colors) against ECCC present weather types recorded on the ground: (**a**) at Eureka; and (**b**) for all 26 stations across Canada. The bars closest to the x-axis represent CloudSat POD for each ECCC weather type. The sample size for each type is given in Table 4, and the POD for a given type is estimated only if its sample size is >20. The error bars represent 95% confidence intervals on each estimate, assuming that the samples are *i.i.d.*

**Table 4.** Table showing the frequency of present weather types (sample size) considered in this study obtained after matching CloudSat and ECCC weather data at Eureka and pan-Canada.

| Present Weather Type | Pan-Canada | Eureka |
|---|---|---|
| Mostly cloudy | 4125 | 794 |
| Mainly clear | 3265 | 586 |
| Clear | 2982 | 725 |
| Snow | 2508 | 484 |
| Cloudy | 2314 | 364 |
| Rain | 1099 | 74 |
| Drizzle | 250 | 5 |
| Thunderstorms | 52 | 0 |
| Snow grains | 50 | 18 |
| Freezing drizzle | 41 | 2 |
| Moderate snow | 40 | 0 |
| Moderate rain | 27 | 0 |

Studies by Hudak et al. [33] and Wang et al. [52] both suggested that details of the CloudSat retrieval may make the instrument prone to incorrectly detecting virga events—where precipitating hydrometeors evaporate or sublimate in the atmospheric boundary layer—as precipitation reaching the surface. This may be related to the absence of retrievals in the 'blind-zone' (Section 2.2), the high sensitivity of the CPR instrument and/or to the reflectivity threshold used in the retrieval algorithm that may detect non-precipitating clouds as precipitation (i.e., false alarms) [33]. Here, we make use of our database of collocated measurements from CloudSat, POSS and in situ observations at Eureka to estimate the POD of virga events there. We define virga events as overpasses where CloudSat indicates precipitation (liquid or solid) reaching the surface, but the ECCC ground observation *and* the ground-based POSS radar both record no precipitation. The rationale is that CloudSat's surface precipitation rate is estimated from radar reflectivity at ~1500 m above the surface, whereas the ground observation and the upward-looking POSS radar measure falling precipitation in the lowest few metres above the surface [45]. Similar to the CPR instrument, the X-band POSS radar is sensitive to low-intensity precipitation, and thus, if both the ECCC ground observation *and* POSS show no precipitation, we have high confidence that precipitation is *not* occurring at the surface, and instead may have evaporated or sublimated in the boundary layer. Following this approach, we identify 221 virga events at Eureka, which represents approximately 9% of the 2378 over-passes where CloudSat incorrectly detected precipitation. We note that spatiotemporal mismatches in sampling between CloudSat and the ECCC ground station likely make this an underestimate of the true frequency of virga events at Eureka, which may explain—at least in part—our lower value than the estimate of ~20% found by Wang et al. [52] across the entire Arctic.

## 4. Validation Across Canada

### 4.1. Detection of Precipitation Occurrence and Phase

Extending the validation to all 26 ECCC weather stations (Figure 1) shows that the values of POD, FAR and HSS across Canada are broadly consistent with those at Eureka (Figure 6). Averaged across all stations, the mean POD, FAR and HSS computed for precipitating and non-precipitating (in parenthesis) conditions are 65.5% $\pm$ 4.3 (90.6% $\pm$ 1.4), 31.4% $\pm$ 5.1 (11% $\pm$ 2.5) and 0.56% $\pm$ 0.04, respectively (Figure 3b), with the uncertainty range representing the 95% confidence intervals for the mean. Despite some variation between the stations, these intervals reflect a high degree of confidence that CloudSat is able to correctly detect the occurrence of precipitation 6–7 times out of 10 across a wide variety of climate zones in Canada. The ECCC weather stations span 37° of latitude and 80° of longitude (Table 3), and thus it is reasonable to ask whether POD varies as a function of spatial location. There appears to be a systematic effect of latitude, whereby

the stations located below 50° N show higher POD, lower FAR, and higher skill scores than the stations above 50° N (Figure 6a,b). In contrast, we do not find any effect of longitude on the POD (estimated visually by comparing the POD for stations at similar latitudes in Figure 1). It seems likely that the influence of latitude relates to a physical property of the hydrometeors—driven perhaps by higher relative humidity at more southerly locations—rather than due to errors introduced by limitations in sampling, because the density of CloudSat overpasses is actually much higher at the northern locations (Table 3). It is beyond the scope of this article to determine the precise physical explanation for this latitudinal effect, but we do highlight the potential significance of this result as it pertains to using CloudSat for snowfall monitoring.

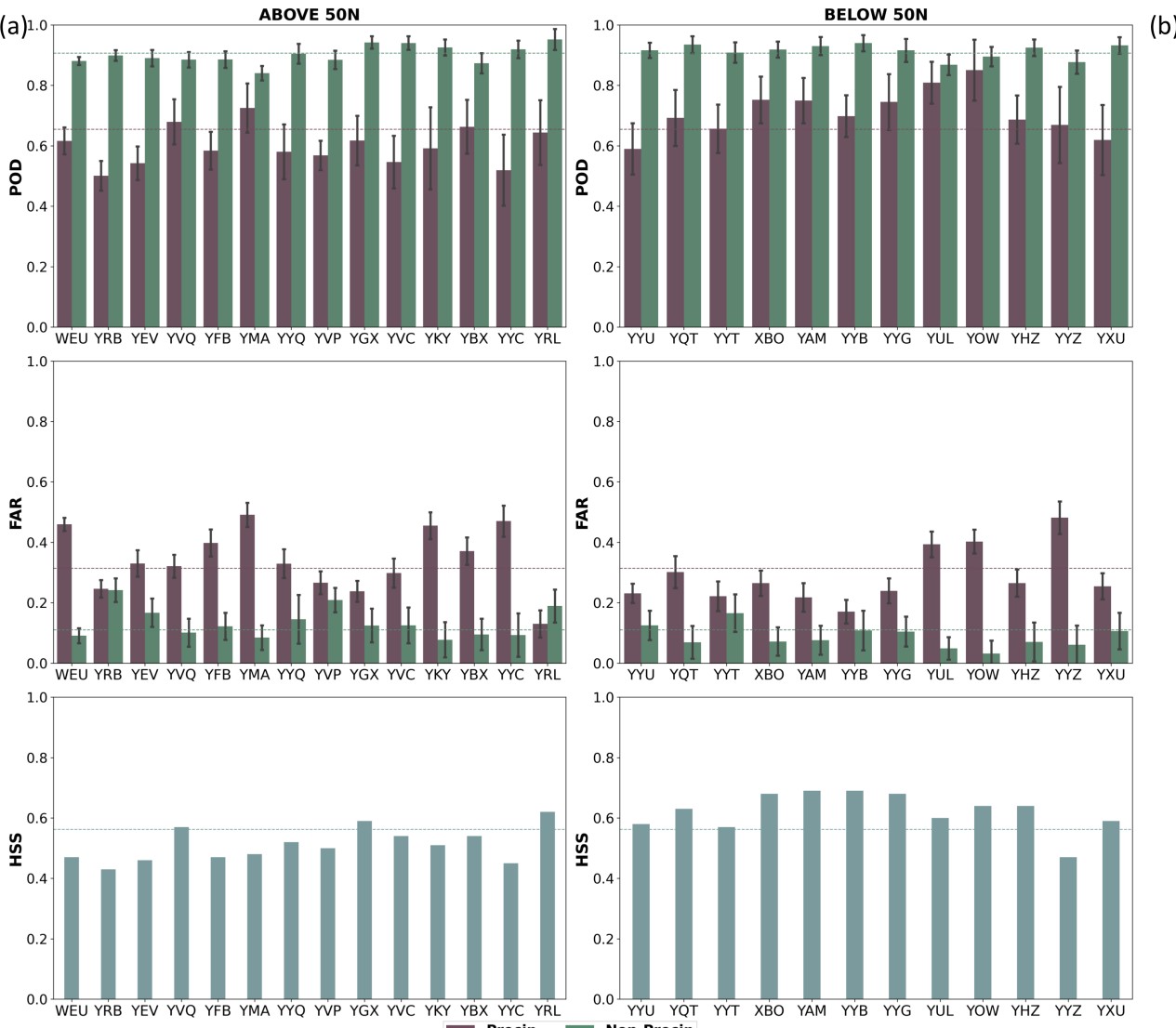

**Figure 6.** Same as the leftmost column in Figure 3, except here each metric is shown for all 26 ECCC weather stations (Table 3). The stations are grouped by latitude: (**a**) all stations >50° N; and (**b**) all stations <50° N.

Turning to the detection of precipitation phase across Canada, the results are generally similar to Eureka. The POD for solid precipitation is uniformly high, exceeding 80% (Figure 7a,b). The POD for liquid precipitation is more variable between stations and tends to be lower at the colder northern locations (e.g., Eureka) that experience a relatively small percentage of rainy days. The FAR is below 10% and the HSS above 0.8 at all but a handful of stations, indicating that CloudSat is able to discriminate between snow and rain with a high degree of accuracy. We again assess the impact of nighttime conditions on POD by

repeating the analysis from Section 3.2, where coincident CloudSat–ECCC observations at each station are separated into summer (day) and winter (night). The findings (not shown) reveal broadly the same conclusion across Canada as for Eureka, which is that the day–night cycle does not appear to make it more challenging for human observers to detect precipitation phase at night, and therefore does not introduce a bias in POD.

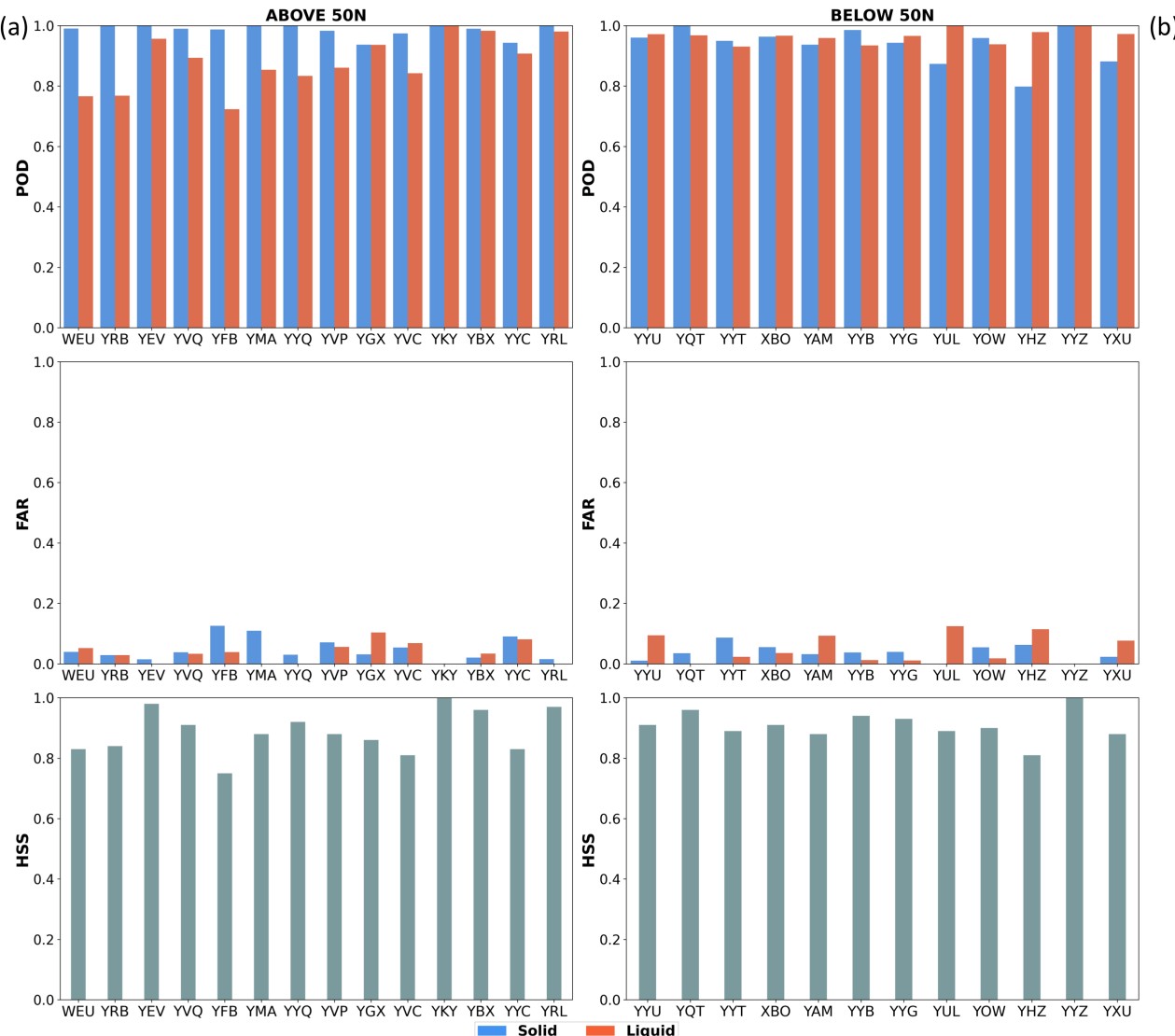

**Figure 7.** Same as the third column from the left in Figure 3, except here each metric is shown for all 26 ECCC weather stations (Table 3). The stations are grouped by latitude: (**a**) all stations >50° N; and (**b**) all stations <50° N.

The pan-Canada POD for precipitation occurrence stratified by cloud cover is again broadly consistent with those for Eureka (Figure 5b), showing values around 80% under full cloud and increasing to >95% under clear skies. For specific weather types, the pan-Canada results show very high accuracy (>90%) for all solid precipitation types and for the most common liquid precipitation types (thunderstorms, rain and moderate rain). The only weather types that have a lower classification rate by CloudSat are drizzle (85%) and freezing drizzle (50%). The latter is based on a relatively small sample of events ($n = 41$), but even considering the larger uncertainty in the POD, freezing drizzle appears to pose the most significant challenge for CloudSat's algorithm. This result makes intuitive sense, because freezing drizzle is, by definition, at the interface of solid and liquid precipitation, involving a complex vertical temperature structure. ECCC defines freezing drizzle as a liquid precipitation type [10], because the hydrometeors are not frozen before they reach

the surface. However, in most freezing drizzle/rain situations, frozen hydrometeors aloft undergo melting within, or just above, the boundary layer, before freezing again on contact with the surface [53]. Therefore, it is entirely conceivable that in some situations where freezing drizzle is observed at the surface CloudSat correctly detects solid precipitation occurring above the boundary layer. In conclusion, it appears that CloudSat's precipitation phase classification exhibits very high accuracy the vast majority of situations, across a wide variety of weather and climate zones.

### 4.2. Influence of Precipitation Intensity

Motivated by our interest in the sensitivity of CloudSat-CPR to light precipitation, we next investigate whether the POD for snow varies as a function of snowfall intensity. We begin by identifying all coincident CloudSat–ECCC observations across Canada with nonzero surface snowfall rates (units mm/h) as estimated by CloudSat. Next, we divide this set of 2374 overpasses into eight bins by their snowfall rate and recompute the POD separately for each group. Figure 8a shows that POD for precipitation occurrence varies considerably as a function of increasing snowfall intensity, from a minimum of 50% for the lightest snowfall rates (<0.01 mm/h) to over 80% for heavier snowfall rates of >0.5 mm/h. Chen et al. [34] reported a broadly similar POD from CloudSat of 76% for very heavy snowfall rates of 1–2.5 mm/h over the Contiguous United States (CONUS). Light snowfall events (<0.05 mm/h) comprise more than 60% of all overpasses in our database of matched observations, which is partly due to CloudSat oversampling Arctic and sub-Arctic locations, where a climatologically drier atmosphere gives rise to lighter precipitation generally, relative to locations in milder, wetter climates at lower latitudes. We find (not shown) that the proportion of snowfall events classified as light declines from ~66% for stations above 50° N, to ~43% for stations below 50° N (Figure 1). Turning to precipitation phase, Figure 8b shows uniformly high accuracy (>90%) for correctly classifying snowfall events as solid precipitation.

Despite the lower POD for drizzle and freezing drizzle in Figure 5b, the result here confirms that lighter precipitation intensity per se is not a factor in misclassifying the precipitation phase. We assume that the sharp decrease in POD of occurrence for light snowfall events results from a difference in the sensitivity between the human observer and the CloudSat-CPR instrument. In general, electronic sensors have far higher sensitivity than human eyes, and so an increased number of missed detections would be expected for low-intensity precipitation [54]. High sensitivity represents a distinct advantage for snowfall monitoring from satellites, particularly in high-latitude locations where snowfall rates are mostly light; for example, the lowest estimated snowfall rate in the 12-year CloudSat record is ~0.003 mm/h, measured near Mayo, Yukon (63.6°N). Except in a few overpasses with spatiotemporal mismatches between CloudSat and ECCC weather data (Section 2.4), it seems highly unlikely for a trained human observer to miss heavy snowfall occurrences. Further analysis suggests that the assessment of CloudSat's POD to changes in precipitation intensity is independent of the day–night cycle, and instrument vs. human sensitivity is a much more likely explanation for the results in Figure 8.

Finally, although the variation of POD with rainfall rate cannot be estimated because CloudSat does not provide a rain rate product over land [55], we can estimate the frequency of overpasses where the ECCC ground observation indicates liquid precipitation but Cloud-Sat classifies solid precipitation. This occurs in a small (~5%) number of cases, and always when the near-surface air temperature is close to freezing. These cases are undoubtedly related to the temperature threshold used to partition solid and liquid precipitation in the CloudSat 2C-PRECIP-COLUMN product [37], on which the 2C-SNOW-PROFILE surface snowfall rate product is based [46]. The mean near-surface air temperature for these misclassified overpasses is 1.6 °C, within the 0–2 °C range in which 2C-SNOW-PROFILE uses the melted fraction to inform its classification of solid or liquid precipitation.

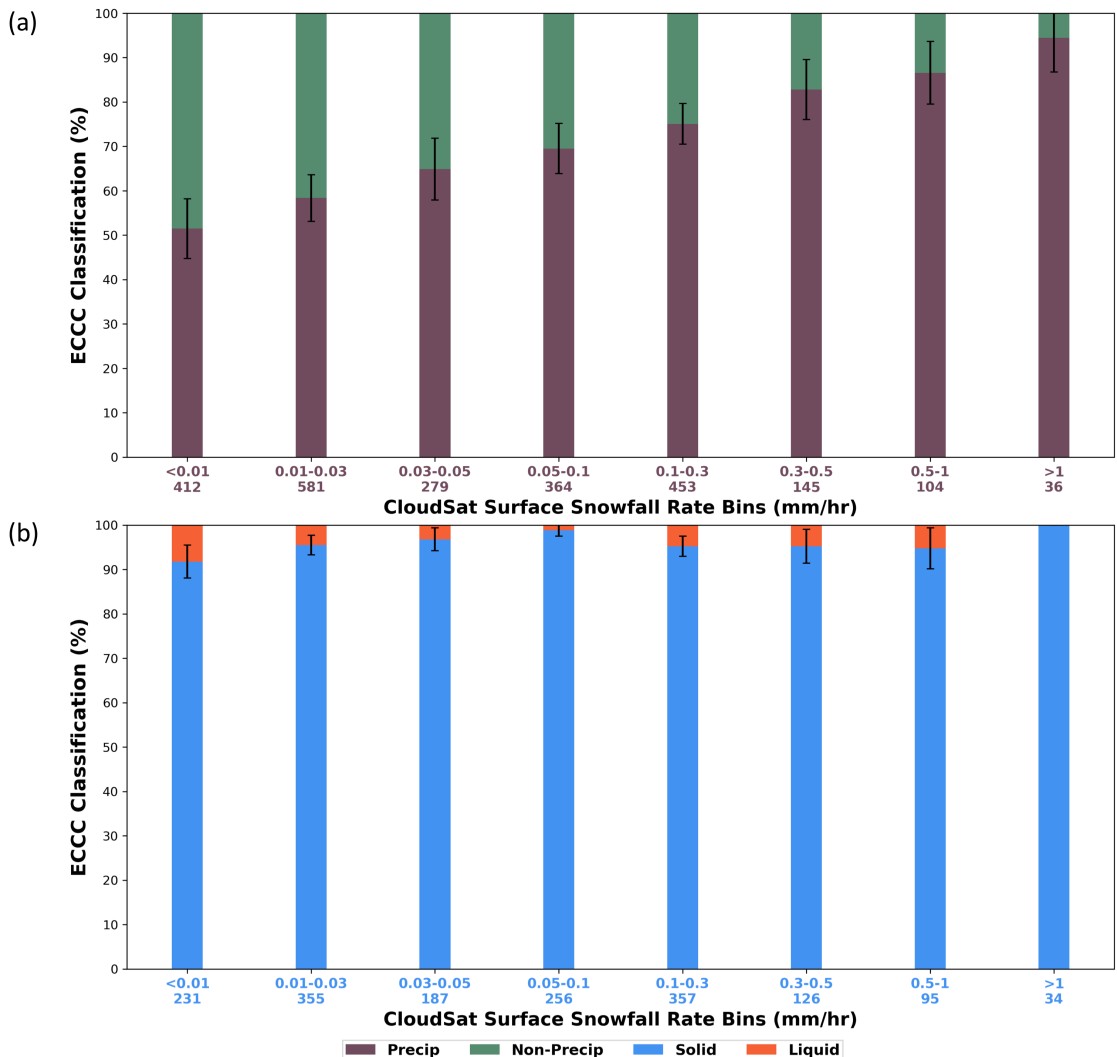

**Figure 8.** Percentages of CloudSat solid precipitation (**a**) occurrence and (**b**) phase that were reported in the ECCC ground data, binned using the CloudSat snowfall rate from 2C-SNOW-PROFILE. The number of coincident observations in each bin is shown under the x-axis below the bin intensity. The error bars represent 95% confidence intervals on each estimate, assuming that the samples are *i.i.d.*

### 4.3. Physical Factors Affecting Detection

In Section 4.1, we present higher rates of missed detection of precipitation occurrence than misclassification of precipitation phase. To understand the physical factors that influence CloudSat's missed detections, we now examine coincident measurements of near-surface reflectivity, near-surface temperature and the height of the top of the lowest significant cloud layer, which are known to exert important controls on the retrieval accuracy of CloudSat [29,33]. We examine the distribution of these three variables as retrieved by CloudSat and interpolated from the ECMWF-AUX auxiliary dataset, for overpasses where the ECCC station recorded non-precipitating or precipitating conditions (Figure 9).

As expected, higher (lower) reflectivity from CloudSat coincides with the satellite detecting (no) precipitation; however, the reflectivity is noticeably higher for the overpasses where CloudSat detects no precipitation and ECCC (the ground truth) detects precipitation (green boxes in Figure 9a). There are two types of missed detection, and each has different characteristics that hint toward a physical explanation. First, there are 1145 overpasses where ECCC detects no precipitation and CloudSat detects precipitation (lefthand aubergine boxes in Figure 9). Roughly 90% of these overpasses have reflectivity above

the −15 dBZ threshold for precipitation detection by CloudSat [51], which suggests that CloudSat has encountered a significant cloud layer that is likely to be producing precipitation at the cloud-layer height, but it does not reach the surface (i.e., virga). The fact that these 1145 missed detections represent 9% of our total sample of 12,686 non-precipitating overpasses across Canada, agrees precisely with our independently-derived estimate of the frequency of virga at Eureka (Section 3.2). This provides high confidence that the missed detection of non-precipitation by CloudSat is almost entirely explained by the presence of the CloudSat blind-zone, and the reliance on extrapolating the precipitation occurrence at 1.5 km altitude down to the surface.

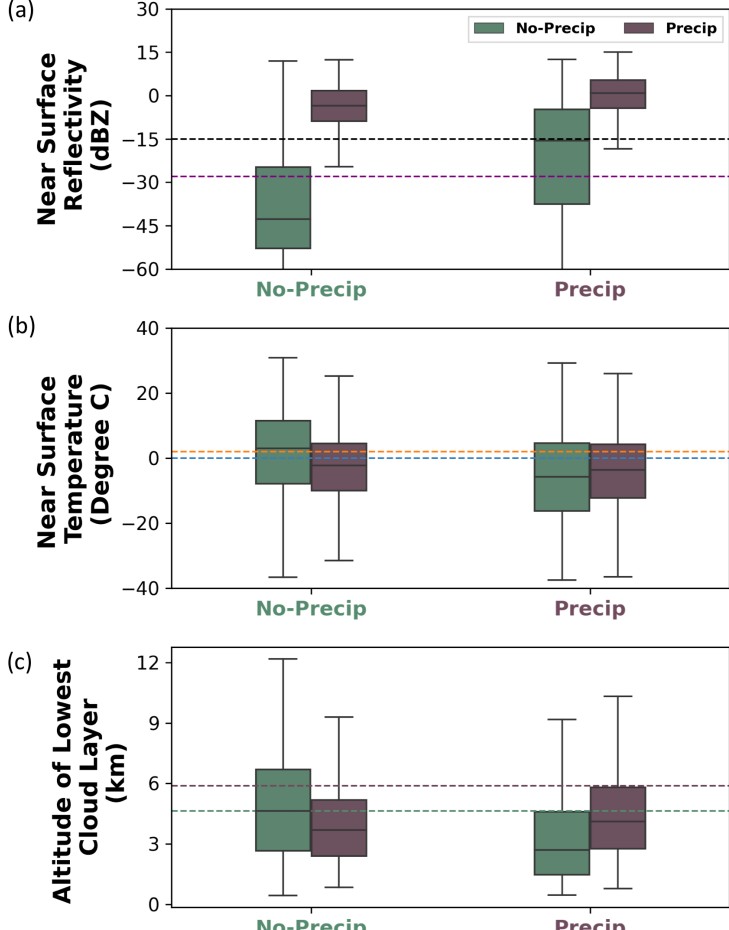

**Figure 9.** Box plots showing the distributions of: (**a**) CloudSat reflectivity (dBZ); (**b**) surface temperature (°C); and (**c**) altitude of the top of the lowest significant cloud layer (km) for precipitating and non-precipitating weather types recorded at ECCC ground stations across Canada for precipitation occurrence. The dotted lines in (**a**) represent the minimum detectability limit of CloudSat-CPR (−28 dBZe) and the reflectivity threshold estimated by Haynes et al. [51] for precipitation occurrence (−15 dBZe). Near-surface reflectivity values <−60 dBZe (far below the threshold to be detected as precipitation) are not shown in the figure. The dotted lines in (**b**) represent the temperature thresholds used in the CloudSat precipitation phase identification algorithm for classifying falling precipitation as snow (0 °C) or rain (2 °C) [37]. The dotted lines in (**c**) show the mean altitude of cloud tops estimated for the overpasses where the CloudSat precipitation phases agreed with ECCC weather data.

The second group is the 1443 overpasses where the ground observation detects precipitation and CloudSat detects no precipitation. These overpasses have a wide range of reflectivity, but they tend to be associated with a lower cloud layer than overpasses with correctly-detected precipitation (righthand green box in Figure 9c). This group divides

roughly equally into a set of 702 overpasses with reflectivity above CloudSat's threshold for precipitation detection (−15 dBZ), and a set of 741 overpasses with reflectivity less than or equal to threshold. The overpasses with higher reflectivity (>−15 dBZ) tend to be located closer to the station than those with lower reflectivity (≤−15 dBZ) (Figure 10a), suggesting that spatial sampling is not a primary driver of their missed detection. In addition, a much larger proportion of the high reflectivity overpasses exhibit cloud layer heights below 3 km (Figure 10c). This evidence leads us to conclude that the overpasses with high reflectivity may be examples of shallow cumuliform precipitation [30], and because more than 65% have surface temperatures below freezing (Figure 9b) these are predominantly snowfall events. Finally, the missed detections with low reflectivity tend to be more than 50 km from the station, and with a wider distribution of cloud top heights (Figure 10a,c). Therefore, we conclude that these are likely overpasses where precipitation—possibly on a small scale—is occurring at the station, and CloudSat passes sufficiently far from the station that it detects only a non-precipitating cloud layer, or clear sky.

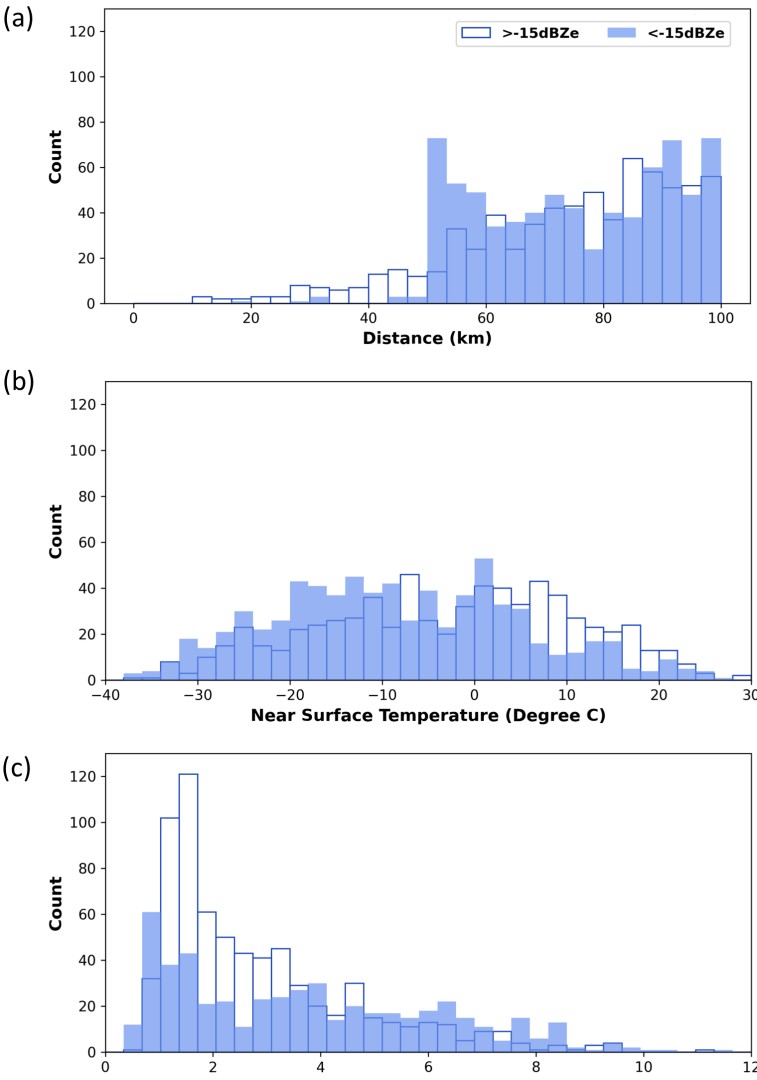

**Figure 10.** Histogram showing the distributions of: (**a**) the mean distance of each overpass from the ECCC station (km); (**b**) near surface temperature (°C); and (**c**) altitude of the top of the lowest significant cloud layer (km) for the overpasses where CloudSat detects no precipitation and the ECCC ground observation detects precipitation. The missed detections are divided into two groups: one with reflectivity ≤−15 dBZ (blue filled bars) and one with reflectivity >−15 dBZ (hollow bars).

## 5. Summary and Conclusions

This presents presents a validation of CloudSat-CPR's precipitation occurrence and phase retrieval over Canada against in situ present weather data recorded by human observers. Both at the highly-sampled Eureka weather station and at 26 ECCC weather stations across Canada, CloudSat is able to correctly detect the occurrence of precipitation around 60% of the time and correctly detects non-precipitation around 85% of the time. False-alarms, where CloudSat's algorithm detects precipitation but none was observed on the ground, occur around 45% of the time at Eureka and 30% of the time across Canada. For precipitating events, CloudSat accurately classifies more than 95% of snowfall events and around 85% of rainfall events, with very few false-alarms, indicating that the primary challenge for CloudSat retrievals is to detect whether precipitation is occurring, rather than what the phase is.

Missed detection of precipitation occurrence is more frequent under cloudier skies, in the presence of drizzle, freezing drizzle or lighter precipitation. The high sensitivity of CloudSat enables it to detect very light precipitation (<0.01 mm/h), which often evaporates/sublimates in the boundary layer and does not reach the surface (i.e., virga). Our analysis reveals that approximately 9% of ground observations that are recorded as non-precipitating may, in fact, be virga events, which is somewhat lower than a previous estimate of ~20% across the entire Arctic [52]. Two groups of overpasses are identified where precipitation is observed on the ground, but CloudSat detects no precipitation: one that is likely caused by shallow cumuliform precipitation occurring in the boundary layer (and therefore within CloudSat's blind-zone) and a second resulting from CloudSat's trajectory being sufficiently far from the station that it does not detect the precipitation occurring at the station.

To our knowledge, this study presents the first detailed validation of precipitation occurrence and phase from CloudSat against in situ weather observations across Canada. The precipitation-flag-based validation methodology described in this paper is robust, simple and reproducible and can be applied at all locations and with any other satellite instrument providing adequate precipitation sampling. Despite differences in sensitivity between instruments and human observers, and the spatiotemporal mismatch inherent in the method of collocating observations, this validation method provides a useful basis for studying rain–snow partitioning. We believe that our results provide a potential pathway to leveraging satellite precipitation observations in general to improve precipitation phase representation in weather, climate or hydrologic models for locations with limited in situ observational data. It is well-known that rain–snow partitioning in nature is highly-variable in space and time [13,56–58]. However, our results suggest that CloudSat's retrieval algorithm is highly accurate at detecting precipitation phase and we recommend increased use of this product, possibly to aid development and/or validation of rain–snow partitioning in atmospheric or land surface models [59]. Moreover, CloudSat precipitation products, together with their auxiliary meteorological information, represent a large quality-controlled data archive, which can be used to develop and train new data-driven phase partitioning schemes, perhaps using machine-learning methods that require large sample sizes [60].

**Author Contributions:** Conceptualization, R.K. and C.G.F.; methodology, R.K. and C.G.F.; software, R.K.; formal analysis, R.K. and C.G.F.; data curation, R.K.; writing—original draft preparation, R.K.; writing—review and editing, R.K. and C.G.F.; visualization, R.K.; supervision, C.G.F.; and funding acquisition, C.G.F. All authors have read and agreed to the published version of the manuscript.

**Funding:** This research was funded by the Canadian Space Agency Earth System Science: Data Analyses Fund, grant number 16SAUSSNOW.

**Institutional Review Board Statement:** Not applicable.

**Informed Consent Statement:** Not applicable.

**Data Availability Statement:** Data and code are available on request from the corresponding author.

**Acknowledgments:** We thank Peter Rodriguez for providing the POSS sensor data and Paul Kushner, Claude Duguay and Wesley Van Wychen for their helpful comments on earlier versions of these results.

**Conflicts of Interest:** The authors declare no conflict of interest.

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
