# Peer review of "Validation of CloudSat-CPR Derived Precipitation Occurrence and Phase Estimates across Canada"

_atmosphere, doi:10.3390/atmos12030295_

Round 1

Reviewer 1 Report

Review comments for the manuscript: “Validation of CloudSat-CPR Derived Precipitation Phase Estimates Across Canada”, by Kodamana and Fletcher.

The manuscript has been substantially revised by the authors and it looks now very clear in its message and methodology. I think the authors did a very good job addressing my concerns, especially the statistical score and the distance from the stations ones.

I suggest some minor comments to make some concepts a bit more clear. I left previous comments to better contextualize, new comments start with ***.

l.84: why only 27 stations if there are 1735 ECCC weather stations in Canada?

Response: These 26 ECCC stations were the only ones where there was both availability of CloudSat retrievals within the selected spatial radius (100km), and a sufficiently large sample of coincident ECCC hourly weather data to make the validation feasible.

*** I suggest to add this in the text, simply stating that you consider 26 stations that have overpasses within 100km and enough sampling.

l.193: My first impression looking at fig. 3c is that it does not provide an immediate insight on the temperature profile associated with each vertical bin or timeframe. What about plotting the actual temperature profile instead of the mean temperatures? This way we can see what happens over the ‘liquid’ sections compared to the ‘solid’ sections of the orbit. Maybe (and this is just a suggestion) also a precipitation type flag would help following what is going on, something like the blue/orange/green colorcode of figure 3a but along the vertical profile.

Response: We replaced the mean temperature in Fig 3c. with the time series of near surface temperature. The near surface temperature profile together with reflectivity profile in Fig. 3b help to understand the changes in precipitation flags (from solid to liquid and vice versa).

*** In the new version fig.3 becomes fig.2: I definitely like the temperature profile better. I was looking at the snowfall rate profile panel (fig.2d) and I noticed some mismatch between the profile and the orbit classification in fig.2a. Is this the entire portion of the orbit highlighted in blue, orange and green in fig.2a or just a part of it? I can see in the vertical profile (fig. 2d) the gap corresponding to the first green (no-precip) and then the larger green+orange and finally the small orange gap, but I don’t see the last large orange gap here.

Section 3.2: if you look for a reason for the 20-30% mismatch on phase classification, I could see here two main things: 1) because of the blind zone, I would like to see if a first precipitation vs no precipitation classification would help understanding what is going on. I think the mismatch between the non precipitating and the precipitating events has different nature and needs to be analyzed separately. 2) The fact that you assign to the entire overpass one phase based on the most frequent phase could be a bit tricky because let’s say for example that at the very end/beginning of the overpass you have a lot of raining profiles because at lower latitude and elevation for example and then you have less snowing profiles closer to the station, probably the liquid classification for that overpass is not very accurate. I would suggest trying to classify the overpass phase based on frequency but also weighting the profiles based on the vicinity to the station, maybe some of the misclassification could be avoided. Same thing for the non precipitating profiles, probably the “at least one profile >0” is a too stringent condition if that only profile is at the very edge of the overpass.

Response: We updated the CloudSat overpass classification methodology to incorporate the distance of profiles from stations under consideration. Each profile is inversely weighted based on its distance from station and a thresholding system is used to classify overpasses into precipitating or non-precipitating at first. Phase classification (solid/liquid) is validated for the overpasses classified as precipitating by the current methodology. The thresholding system removes the stringent condition previously used to estimate overpass precipitation phase. The improvement in skill scores brought about by the updated methodology over the previous (Default) method is shown Fig. 4.  

*** The new method for classifying the overpass definitely improves the results. I am wondering if it is still necessary to keep the Default results in the paper.

l.383: “This strongly suggests that false negative misclassifications are associated with shallow precipitation generated within the boundary layer”: but you tested this with POSS (l.242) and verified that shallow precipitation does not affect misclassification (at least at Eureka that is anyway very representative of the pan-Canada on average).

Response: We thank the reviewer for highlighting this apparent contradiction. The revised text (Section 4.3) goes into greater detail in separating out cases of misclassification, and our analysis suggests that shallow precipitation may be associated with a smaller subset of false negative cases, where CloudSat detects relatively high reflectivity (> -15dBZ) but still classifies no precipitation.

*** l.432-435: I am probably missing something here, but if you could determine the cloud base height it means that it is still above the blind zone, so why CloudSat doesn’t detect precipitation? If the near surface bin has reflectivity >-15dBZ, the temperature is below freezing, I don’t understand why CloudSat does not detect shallow convective here. Usually shallow convective is difficult to detect when the top of the cloud is below 1.5km (so very shallow), but 1.5-3km is actually a good range for detecting shallow (see Kulie et al. 2016 fig.1). Any explanation for that?

Figure 5: do the dashed lines represent the mean values? Are they separated from >50 to <50 latitude or is this a total mean value?

Response: The dashed lines represent pan-Canada mean values. The caption has been updated to clarify this information.

*** Figure 5 is now figure 6: I think the caption is not correct, it is a copy of fig.7 caption.

Other comments:

*** Reading the manuscript it took me a while to figure out that you were using ‘misclassification’ interchangeably for precip vs no-precip and liquid vs solid. The first thing I would think about in a paper like this when mentioning misclassification is the liquid vs solid (so phase classification or misclassification). For a better understanding I would suggest to reword throughout the manuscript the misclassification of precip vs no-precip with ‘detection/miss’ (for example title of section 3.2 could be ‘factors influencing detection’) and keep misclassification for the phase to avoid misinterpretation of the results.

*** When you mention cases, observations, orbits, do you refer to the same thing, i.e. a single overpass within 100 km from the station?

*** l.313: do you mean ‘underestimate’?

*** l.377: is this 70N or did you mean 50N?

Reviewer 2 Report

I appreciate the authors’ great efforts on the revisions compared to the original manuscript and to address my previous concerns. The current revision is more readable and shows a significant improvement.

However, I find the authors’ definition on POD and FAR difficult to follow, particularly FAR. Typically, based on Table 2 (assume solid-> precipitating and liquid->non-precipitating), one would define as follows:

POD = (a+d)/(a+b+c+d)

          or

POD_prec = a/(a+b+c+d) and POD_nonprec=d/(a+b+c+d)

FAR = b/(a+b+c+d)

Missed = c/(a+b+c+d), which does not have to be shown.

I recommend the authors to clarify this before acceptance for publication.

Round 2

Reviewer 2 Report

Recommend for publication

This manuscript is a resubmission of an earlier submission. The following is a list of the peer review reports and author responses from that submission.

Round 1

Reviewer 1 Report

Review of the Manuscript: “Validation of Precipitation Phase Estimates Across Canada from CloudSat-CPR”, by Kodamana and Fletcher.

The authors validate CloudSat precipitation phase information against ground based weather stations over Canada. The work presented here is a systematic and straightforward analysis about the various factors affecting the different phase classification made by a satellite-based radar, very sensitive to small hydrometeors, with a blind zone below about 1200m and local human observations less sensitive to very small hydrometeors but with the possibility of observing weather conditions at ground level. After presenting a case study over Eureka station and a long term statistical analysis (about 10 years) over the same station, the authors extend the results to the pan-Canada region showing good performances especially on classifying solid precipitation.

As already mentioned, the analysis is systematic, the authors considered many factors influencing the phase classification itself and the comparison between the two different measurement systems. Phase classification is one of the major issues in precipitation remote sensing and there is an extreme need to improve how algorithms and models treat it. This work is therefore important to establish what the main issues are and with stronger conclusions could lead to a systematic way to determine uncertainties. The only big concern that I have is that for the statistical analysis only the POD index has been used. This kind of index only considers events that are correctly classified and a random component is also included. Usually this index goes together at least with the FAR index to get a sense of the false alarms that you also advertised to be a possible issue given the methodology used to define non precipitating overpasses. POD and FAR are not strictly complementary so the information you provide about the misclassification does not really give the same complete statistical view. There could be very good POD values but if the FAR is high as well, the methodology used maybe is not so good as it could appear looking only at one statistical parameter. Also, having more than 2 classes, some multi-classes parameters could be considered, like KSS or HSS for example. For this lack of completeness in the statistical analysis I suggest major revisions before publication. I leave to the authors to find the best way to complete the statistical analysis and pick the scores more suitable for the proposed work. More strong conclusions suggesting how the results could be considered for determining phase classification uncertainties would also improve the overall quality and message of the paper.

Other minor comments are listed below:

l.97-100: I see some confusion in the CloudSat products description: the 2B-GEOPROF product doesn't really identify the occurrence of precipitation but when the radar return power is likely to be due to scattering by clouds or other hydrometeors and when it is likely to contain only noise. It substantially identify hydrometeors. And I don’t really see any reference to ‘significant cloud’ in the literature but more ‘significant echo’. Precipitation and so the occurrence of precipitation comes into play with the 2C-PRECIP-COLUMNS product that actually ‘decides’ what is going on (precipitation wise).

l.81: why only 27 stations if there are 1735 ECCC weather stations in Canada?

l.148-151: are you doing the phase assignment of mixed precipitation based on the melted mass fraction or do you leave this to the algorithm? It is not clear to me why you have to take care of this if the algorithm, as you state, already does it.

l.193: My first impression looking at fig. 3c is that it does not provide an immediate insight on the temperature profile associated with each vertical bin or timeframe. What about plotting the actual temperature profile instead of the mean temperatures? This way we can see what happens over the ‘liquid’ sections compared to the ‘solid’ sections of the orbit. Maybe (and this is just a suggestion) also a precipitation type flag would help following what is going on, something like the blue/orange/green colorcode of figure 3a but along the vertical profile.

l.195: I suggest to specify that the increased reflectivity is at the near surface bin.

Section 3.2: if you look for a reason for the 20-30% mismatch on phase classification, I could see here two main things: 1) because of the blind zone, I would like to see if a first precipitation vs no precipitation classification would help understanding what is going on. I think the mismatch between the non precipitating and the precipitating events has different nature and needs to be analyzed separately. 2) The fact that you assign to the entire overpass one phase based on the most frequent phase could be a bit tricky because let’s say for example that at the very end/beginning of the overpass you have a lot of raining profiles because at lower latitude and elevation for example and then you have less snowing profiles closer to the station, probably the liquid classification for that overpass is not very accurate. I would suggest trying to classify the overpass phase based on frequency but also weighting the profiles based on the vicinity to the station, maybe some of the misclassification could be avoided. Same thing for the non precipitating profiles, probably the “at least one profile >0” is a too stringent condition if that only profile is at the very edge of the overpass.

l.286: You are analyzing the events that are supposedly virga cases at Eureka based on the fact that ECCC weather and POSS agree as no precipitation and CloudSat says it is precipitating (liquid or solid). See my comment before, maybe a weighted frequency of precipitating pixels could help, we are not sure if these overpasses are actually virga cases since the orbit could pass 100 km away and have an only precipitating profile. Also, table 4 contains only POD values, I don’t see any reference to the numbers mentioned for this virga analysis.

l.309: There is a decrease in POD with a DECREASE in diameter.

l.313: “Compared to Eureka, there is a pan-Canada increase in the percentage of solid precipitation cases misclassified as liquid, and a decrease in liquid precipitation misclassified as solid (Table 5).”: I get confused looking at figure 4 because for example for ‘rain’ (we don’t have other liquid classes to compare) the amount of solid misclassification is lower for the same class considering the pan-Canada dataset, while for the solid ‘snow’ category, there is no liquid misclassification according to the histogram. Again, this confusion that I am struggling with could be due to the fact that the POD index is used instead of the actual number of cases, with a contingency table filled with POD values instead of cases numbers. I would suggest to fix these concepts throughout the manuscript.

l.316-317: “a larger number of these events are misclassified as liquid (13.2%) rather than solid (17%)”. Maybe you mean the opposite since the solid percentage (17%) is higher than the liquid one (13.2%)?

l.334-337: and what about precipitation not actually reaching the ground (are here virga cases included?) or overpasses too far from the station so the precipitation system does not actually reach the station itself?

l.383: “This strongly suggests that false negative misclassifications are associated with shallow precipitation generated within the boundary layer”: but you tested this with POSS (l.242) and verified that shallow precipitation does not affect misclassification (at least at Eureka that is anyway very representative of the pan-Canada on average).

l.388: maybe temperature influences those 3.2% (2.6%) of misclassified cases not considered in this last analysis?

Table 2: “co-ordinates: would be “coordinates”.

Table 4 and 5: I am a bit confused by the numbers in these two tables. The label states this is a contingency table so I would expect the number of CloudSat overpasses in each category (or a percentage of them). But it also says that this is the POD calculated between CloudSat and ECCC (POSS). How did you calculate the non-diagonal POD? I am not sure I understand the statistical meaning here, why calculating a POD between two different categories like a POD on the liquid/solid box for example or solid/liquid or nonprecip/liquid etc?

Table 4: no need to repeat that “The POD of CloudSat using POSS data is shown in brackets since you already put the “(in parenthesis)” in the previous sentence.

Figure 1: the 50 N line is there to separate the ground stations but in the end I don’t see this information used for any analysis except for separating figure 5 histogram in 2 rows, but also the mean values seem to be total and not above or below 50N (see comment on figure 5).

Figure 2: please specify that the dashed line refers to the right Y-axis and the columns are the POD values (left Y-axis).

Figure 4: is this a POD again or is this a percentage of classified cases for each category?

Figure 5: do the dashed lines represent the mean values? Are they separated from >50 to <50 latitude or is this a total mean value?

Reviewer 2 Report

See attachment
